# Nonsterile immunity to cryptosporidiosis in infants is associated with mucosal IgA against the sporozoite and protection from malnutrition

**Mamun Kabir**[1], **Masud Alam**[1], **Uma Nayak**[2,3], **Tuhinur Arju**[1], **Biplob Hossain**[1], **Rubaiya Tarannum**[1], **Amena Khatun**[1], **Jennifer A. White**[4], **Jennie Z. Ma**[2,4], **Rashidul Haque**[1], **William A. Petri Jr**[4,5,6]*, **Carol A. Gilchrist**[4]

1 Emerging Infections and Parasitology Laboratory, Infectious Diseases Division, International Centre for Diarrhoeal Diseases Research, Mohakhali, Bangladesh, 2 Department of Public Health Sciences, University of Virginia, Charlottesville, Virginia, United States of America, 3 Center for Public Health Genomics, University of Virginia, Charlottesville, Virginia, United States of America, 4 Department of Medicine, University of Virginia, Charlottesville, Virginia, United States of America, 5 Department of Microbiology, Immunology and Cancer Biology, University of Virginia, Charlottesville, Virginia, United States of America, 6 Department of Pathology, University of Virginia, Charlottesville, Virginia, United States of America

* wap3g@virginia.edu

**Data Availability Statement:** Select clinical metadata is available on the NCBI's dbGaP under accession number phs001665.v2.p1. The data for

## Abstract

We conducted a longitudinal study of cryptosporidiosis from birth to three years of age in an urban slum of Dhaka Bangladesh. Fecal DNA was extracted from monthly surveillance samples and diarrheal stool samples collected from 392 infants from birth to three years. A pan-Cryptosporidium qPCR assay was used to identify sub-clinical and symptomatic cryptosporidiosis. Anthropometric measurements were collected quarterly to assess child nutritional status. 31% (121/392) of children experienced a single and 57% (222/392) multiple infections with *Cryptosporidium*. Repeat infections had a lower burden of parasites in the stool (Cq slope = -1.85; p<0.0001) and were more likely to be sub-clinical (Chi square test for trend; p=0.01). Repeat infections were associated with the development of growth faltering (Pearson correlation = -0.18; p=0.0004). High levels of fecal IgA antibodies against the *Cryptosporidium* Cp23 sporozoite protein at one year of life were associated with a delay in reinfection and amelioration of growth faltering through three years of life (HAZ IgA high responders -1.323 ± 0.932 versus HAZ -1.731 ± 0.984 p=0.0001). We concluded that non-sterile immunity to cryptosporidiosis in young children was associated with high levels of mucosal IgA anti-Cp23 and protection from diarrhea and growth faltering.

**Trial Registration:** NCT02764918.

## Author summary

*Cryptosporidium* is one of the top causes of diarrhea and growth faltering in Bangladesh infants. We discovered that a prior infection resulted in incomplete immunity that

this study is collected as a sub-study of dbGaP phs001475.v2.p1.

**Funding:** This work was supported by grants NIH R01-043596 to WP and CG, R21-142656 to CG and the Bill & Melinda Gates Foundation OPP1100514 to WP. The funders had no role in study design, data collection and analysis or decision to submit for publication.

**Competing interests:** The authors have declared that no competing interests exist.

protected from diarrhea and growth faltering but not infection and was associated with mucosal IgA against a sporozoite surface protein Cp23. The most important implication of these findings is that a cryptosporidiosis vaccine may not need to achieve complete protection from infection to have a beneficial impact on child health.

## Introduction

*Cryptosporidium spp.* parasites are leading causes of diarrheal disease in infants living in low and middle income countries [1–4]. They are additionally a cause of water-borne outbreaks of diarrhea in high income countries and of chronic diarrhea in people living with HIV infection [5]. There is no vaccine and development will require an understanding of the natural history of cryptosporidiosis [3,6–12]. To this end, a community-based prospective cohort study of cryptosporidiosis was begun in 2014 [13]. The study subjects, born in an urban slum in Dhaka, Bangladesh were enrolled during the first week of life [13–15]. In humans and in animal models vaccination or prior infection resulted in partial protection against reinfection [16–19]. For example we observed that high fecal IgA against the sporozoite protein Cp23 delayed but did not prevent a repeat infection with *Cryptosporidium spp.* [20]. Ajjampur et al observed a decrease in the incidence of diarrhea in reinfected children [21]. In contrast Kattula et al found that while the reinfection frequency was decreased the proportion of symptomatic disease was unchanged [9]. In human volunteer studies second infections were associated with reduced parasite burden and less severe diarrhea [22].

In addition to diarrheal disease cryptosporidiosis is associated with development of malnutrition [8,23–26]. Here we report the natural history of cryptosporidiosis from a longitudinal study of urban slum children from birth through three years of age in Dhaka, Bangladesh, demonstrating that immunity is characterized by protection from diarrhea and growth faltering.

## Results

Five hundred children were enrolled within the first week of birth, and of these 392 completed three years of observation. Stool samples were collected monthly and at the time of diarrhea. Successful sample collection and qPCR testing was completed for 96% of monthly surveillance time points and for 84% of the diarrheal cases (Figs 1 and S1 and Table 1). There were 1336 *Cryptosporidium* positive samples for analysis by year 3 (Fig 1). PCR- positive samples were classified as a separate infection if occurring greater than 65 days after the preceding positive sample [13]. The *Cryptosporidium* infection phenotype (diarrheal or sub-clinical) was based upon symptoms at the time of detection of the first *Cryptosporidium* - positive stool sample. Six hundred and ninety eight events met the definition of separate *Cryptosporidium* infections in the 392 children (Table 1).

Of the 698 infections experienced by the 392 infants retained in the study at 3 years of age, 167 were diarrheal and 531 sub-clinical cryptosporidiosis (Table 1). The Cq (cycle of quantification) value of the stool sample in which the parasite was first detected was used as an index of parasite burden.

The slopes derived from the GEE models for sub-clinical (1.9 ± 0.2) and diarrheal (1.49 ± 0.31) infections were not significantly different from each other (S2 Fig). However recurrent infections had as expected a lower amount or burden of *Cryptosporidium* than did the first infection (slope -1.85 ± 0.21; p<0.0001) (Fig 2A) To investigate if a reduction in the severity of clinical disease in recurrent infections could be correlated with the reduced parasite

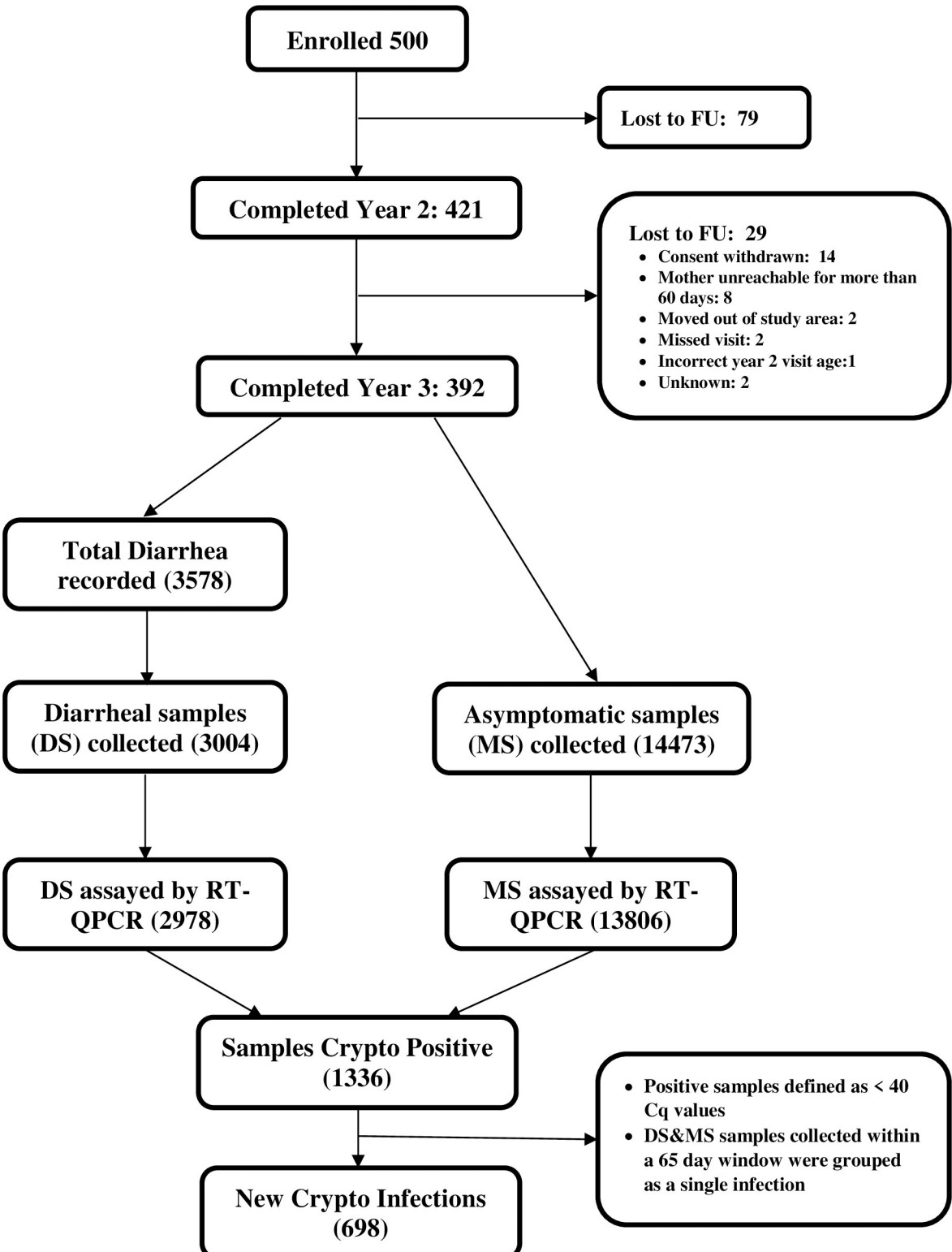

**Fig 1. COHORT diagram.** Study subjects, collected samples and new infection numbers. Abbreviations: RT-qPCR Real-Time quantitative polymerase chain reaction; DS: Diarrheal Stool samples; FU: Follow-up; MS: Monthly Stool Samples, Cq: Cycle Quantification.

**Table 1. Frequency of Diarrheal Cryptosporidiosis in Repeated Infections.**

| Cryptosporidium Infection | Number of infections | Age in days | | Infection phenotype* | | Diarrhea Frequency** | | |
|---|---|---|---|---|---|---|---|---|
| | | Mean ± SD | Range | Diarrhea | Sub-clinical | Mean | Upper limit | Lower limit |
| 1st | 343 | 519 ± 250 | [15-1088] | 96 | 247 | 0.28 | 0.33 | 0.23 |
| 2nd | 222 | 758 ± 209 | [273-1079] | 46 | 176 | 0.21 | 0.27 | 0.16 |
| 3rd | 101 | 892 ±155 | [399-1084] | 17 | 84 | 0.17 | 0.26 | 0.10 |
| 4th | 28 | 939± 129 | [639-1095] | 6 | 22 | 0.21 | 0.41 | 0.08 |
| 5th | 3 | 994 ± 72.5 | [929-1072] | 0 | 3 | 0.0 | 0.70 | 0 |
| 6th | 1 | 1067 | 1067 | 0 | 1 | 0.0 | 0.97 | 0 |

*Chi Square test for trend p=0.011

** The frequency of *Cryptosporidium* associated diarrhea and range of values \by Chi Square test

burden, the amount of sub-clinical and diarrheal disease in recurrent infections was investigated (Table 2).

To investigate if an initial high burden infection provided better protection against future infections with the *Cryptosporidium* parasite, we compared the Cq values of the infections in children who only had one infection in the first three years of life vs the Cq values of the first infection in children that had repeated infections (Fig 2B). The mean Cq values were similar in both cases (single infections: Cq 27.6 ± 5.4: 1st infection of multiples: 28.8 ± 6.0) and significantly lower than that in subsequent second infections where infections were >1 (Cq second infection: 31.5 ± 4.9).

We next evaluated if the lower parasite burden in repeat infections was influenced by the age of the children. As most recurrent *Cryptosporidium* infections occurred in older children, (Table 1) we analyzed a subset of the *Cryptosporidium* positive samples corresponding to the first to fifth infections in children aged between 2.25 and 2.75 years. The parasite burden measured by qPCR remained significantly lower in the recurrent infections (Fig 3). The negative relationship of lower parasite burden with repeated infections was not an artefact of PCR inhibitors in the stool of older children because detection of the Phocine herpesvirus (PhHV) DNA included as internal extraction control [27] was not significantly affected by the number of prior *Cryptosporidium* infections. We concluded that repeat infections had a lower parasite burden.

The duration of diarrheal disease was similar in the first infection and later reinfections (single infection: 4.8 ± 2.9 days; primary infection: 5.5 ± 3.9 days; later infections: 5.2 ± 3.6 days), however, the proportion of the diarrhea-associated *Cryptosporidium* infections decreased in the recurrent infections (Chi-squared test for trend p=0.011) (Table 1). We concluded that the repeated *Cryptosporidium* infections were more likely to be sub-clinical.

## *Cryptosporidium* and growth faltering

Growth faltering (low height for age; HAZ score) was analyzed from the 3 year old children based on the number of *Cryptosporidium spp.* infections [0–3 years] (both diarrhea and sub-clinical) (Table 2 and S3 and S4 Figs). The association between cryptosporidiosis and HAZ score at three years was examined by multiple regression in order to account for the effect of the confounding variables previously identified (Table 3) [13].

*Cryptosporidium* infection was negatively associated with the HAZ score at 3 years after adjusting for birth length-for-age (LAZ) score and maternal weight and education: each *Cryptosporidium* infection reoccurrence resulted in a decrease in HAZ score (Δ 0.12) at 3 years (Table 3). No significant relation was found between malnutrition at birth (LAZ score) and

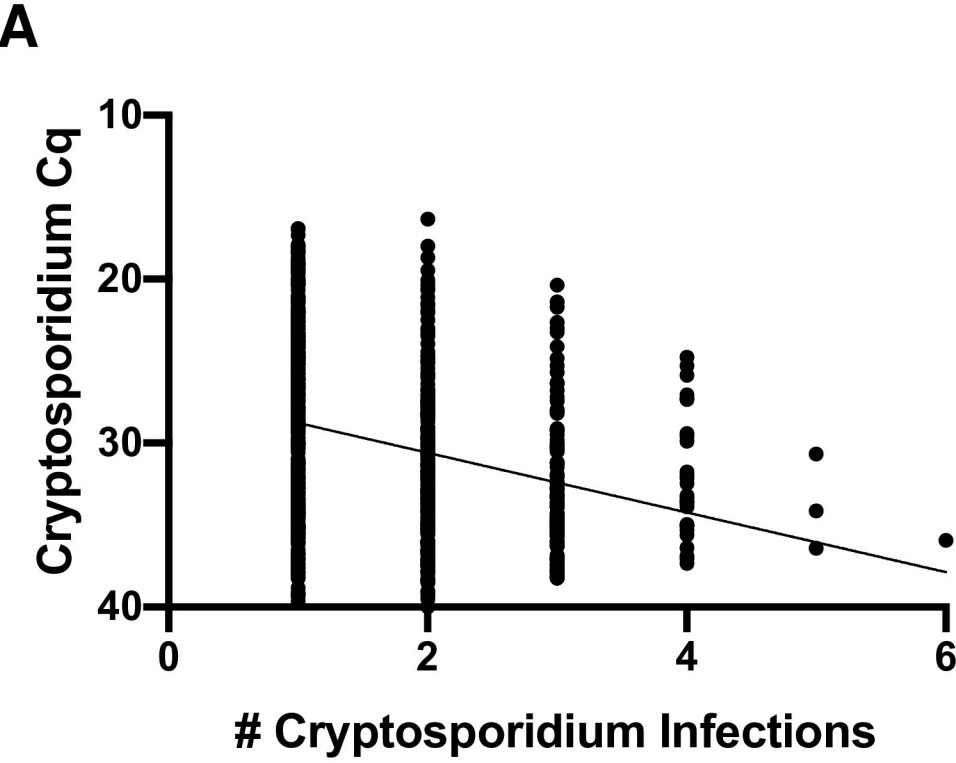

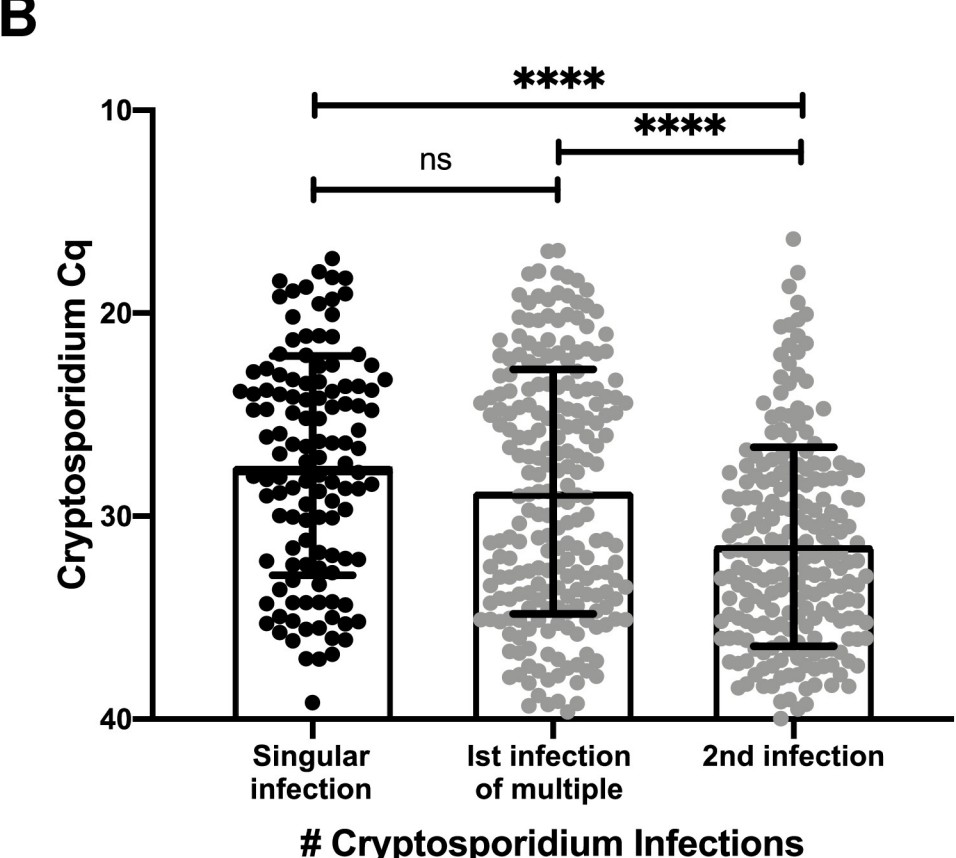

**Fig 2. Parasite burden was lower in recurrent infections. A)** Correlation between parasite burden and the number of *Cryptosporidium* infections. Each symbol represents the first detectable sample of an individual infection. Y-axis represents the quantitative cycle (Cq) of the diagnostic pan-Cryptosporidium PCR assay. X-axis shows the number of *Cryptosporidium* infections that had occurred in this child. The line represents the slope (-1.85 ± 0.21) and Y-intercept (26.95 ± 0.44) estimated from the GEE model with the exchangeable correlation structure (p<0.0001). **B)** Comparison of single infections (black symbol) with those that are part of a series (gray symbol). Bar graph (indicating data mean ± standard deviation) with individual data points. Each symbol on the box plot represents the first positive sample of an individual infection. X-axis refers to Infection number and, if the first infection, whether a second *Cryptosporidium* infection took place in the 3 years of life. Y-axis represents the quantitative cycle (Cq) of the diagnostic pan-Cryptosporidium PCR assay. Horizontal bars represent the result of a non-parametric Kruskal-Wallis test **** indicates p<0.0001.

total number of *Cryptosporidium* infections during the follow-up (Fig 4A). However, the total number of *Cryptosporidium* infections was negatively associated with HAZ score at 3 years (Fig 4B, regression coef=-0.152, p=0.0004) The association between HAZ and *Cryptosporidium spp.* infections was unaffected by whether the event was a sub-clinical infection or diarrheal disease (S5 Fig).

Other measurements that are used as indicators of malnutrition were also significantly associated with the number of *Cryptosporidium* infections. These included mid-upper arm circumference (MUAC) (S6A Fig; MUACZ vs. number of *Cryptosporidium* infections slope: -0.088; p=0.0123 and weight-for-age (WAZ score) (S6B Fig) (linear regression analysis slope: -0.115 p=0.0093). However, neither BAZ (body-mass- for- age) (S6C Fig), used to measure acute protein-energy malnutrition or wasting (WHZ) were affected by a history of *Cryptosporidium* infections (S6D Fig).

The Pearson correlations among the number of *Cryptosporidium* infections, LAZ at birth, diarrheal episodes and HAZ at year 3 confirmed the results of the linear regression analysis (*Cryptosporidium* infections: HAZ at year 3: coef = -0.18, p=0.024; *Cryptosporidium* infections: diarrheal episodes captured (all causes): coef = 0.22, p<0.0001; HAZ at year 3: LAZ at birth: coef = 0.28, p= 0.008). As expected, a significant correlation existed between LAZ at birth and HAZ at year 3 (simple linear regression p<0.0001 Fig 5A).

Enteric pathogens are endemic in the Bangladesh study population [28] and as a consequence, infants enrolled in the study cohort had repeated diarrheal episodes of which only some were associated with infection with the *Cryptosporidium* parasite. However, while *Cryptosporidium* infections (diarrheal and sub-clinical) were significantly associated with child HAZ at year 3 (Pearson's correlation p=0.0004), the number of all-cause diarrheal episodes was not (Fig 5B and Tables 3 and S1). This result supported our conclusion that this growth shortfall was specifically associated with recurrent cryptosporidiosis.

**Table 2. Distribution and Clinical Characterization of repeated *Cryptosporidium* infections.**

| # of Infections | # of Children | Cryptosporidium infections | | | | | | | | | | | | Total # of infections |
|---|---|---|---|---|---|---|---|---|---|---|---|---|---|---|
| | | 1st | | 2nd | | 3rd | | 4th | | 5th | | 6th | | |
| | | DS* | MS** | DS | MS | DS | MS | DS | MS | DS | MS | DS | MS | |
| 0 | 49 | 0 | 0 | 0 | 0 | 0 | 0 | 0 | 0 | 0 | 0 | 0 | 0 | 0 |
| 1 | 121 | **28** | **93** | 0 | 0 | 0 | 0 | 0 | 0 | 0 | 0 | 0 | 0 | **121** |
| 2 | 121 | **34** | **87** | 19 | 102 | 0 | 0 | 0 | 0 | 0 | 0 | 0 | 0 | **242** |
| 3 | 73 | **24** | **49** | 19 | 54 | 15 | 58 | 0 | 0 | 0 | 0 | 0 | 0 | **219** |
| 4 | 25 | **8** | **17** | 7 | 18 | 1 | 24 | 6 | 19 | 0 | 0 | 0 | 0 | **100** |
| 5 | 2 | **1** | **1** | 1 | 1 | 1 | 1 | 0 | 2 | 0 | 2 | 0 | 0 | **10** |
| 6 | 1 | **1** | 0 | 0 | 1 | 0 | 1 | 0 | 1 | 0 | 1 | 0 | 1 | **6** |
| **Total** | **392** | 96 | 247 | 46 | 176 | 17 | 84 | 6 | 22 | 0 | 3 | 0 | 1 | **698** |

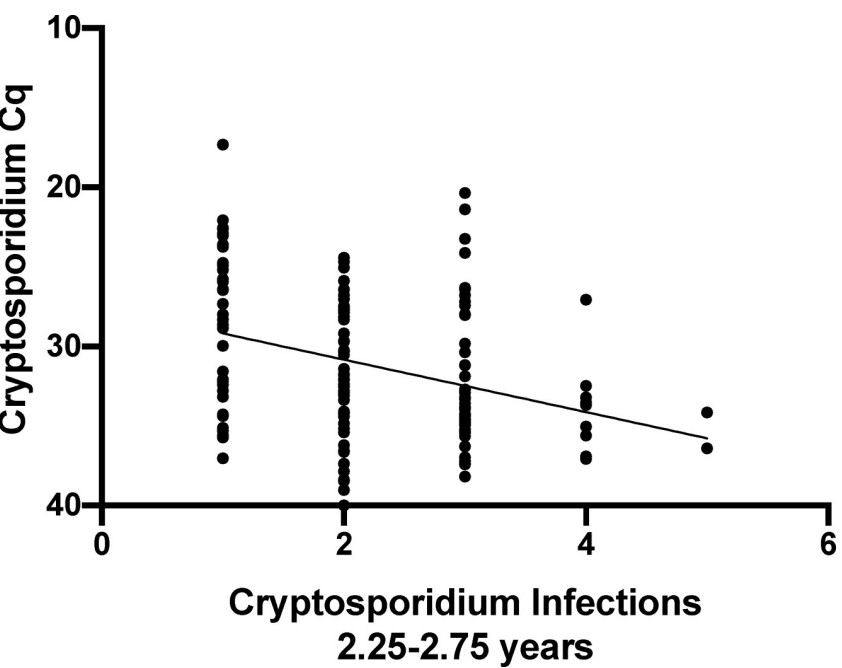

**Fig 3. Parasite burden in older children.** The amount of parasite in stool was determined as a function of the number of *Cryptosporidium* infections in a child by linear regression. The analysis was restricted to children between 2.25 and 2.75 years of age (n=140). Each symbol represents the first detectable sample of an individual infection. Y-axis, quantitative cycle of the diagnostic pan-Cryptosporidium PCR assay (Cq). X-axis, the number *Cryptosporidium* infections that had occurred in each child. Slope: -1.65, R squared value: 0.01125, Significance p<0.0001.

## Mucosal IgA against the sporozoite Cp23 protein was associated with protection from growth faltering

Children with high levels of fecal anti-Cp23 IgA were regarded as being previously exposed to the *Cryptosporidium* parasite even if the parasite had been missed during surveillance for sub-clinical infections. In previous work it was shown that even if *Cryptosporidium* DNA had not been identified high levels (> mean value) of fecal anti-Cp23 IgA at one year of age was associated with an increased resistance to cryptosporidiosis through age three [14,20]. In children with high fecal anti-Cp23 IgA there was an increase in the number of days until a child was reinfected and a subsequent decrease in the proportion of children in this group reinfected during years 2-3 (in children with high fecal anti-Cp23 IgA 77.8% were reinfected versus 92.2%)

Here we additionally discovered that children with high levels (upper 50[th] percentile) of fecal anti-Cp23 IgA at one year of age were protected from growth faltering through year 3

**Table 3. Regression Analysis using selected predictors to test the association of Cryptosporidiosis with Height-for-Age Scores at 3 Years.**

| Parameter | Effect (95% Confidence Interval) | P Value |
|---|---|---|
| Cryptosporidium Infections | -0.12 (-0.197, -0.043) | 0.0024 |
| Child LAZ at Birth | 0.252 (0.159, 0.345) | <0.0001 |
| Maternal Weight | 0.017 (0.007, 0.027) | 0.0011 |
| Maternal Height | 0.037 (0.018, 0.055) | <0.0001 |
| Maternal Education | 0.237 (0.027, 0.448) | 0.0273 |
| Household income | 0.001 (0.000, 0.002) | 0.1527 |
| Treated water | 0.163 (-0.045, 0.371) | 0.1234 |

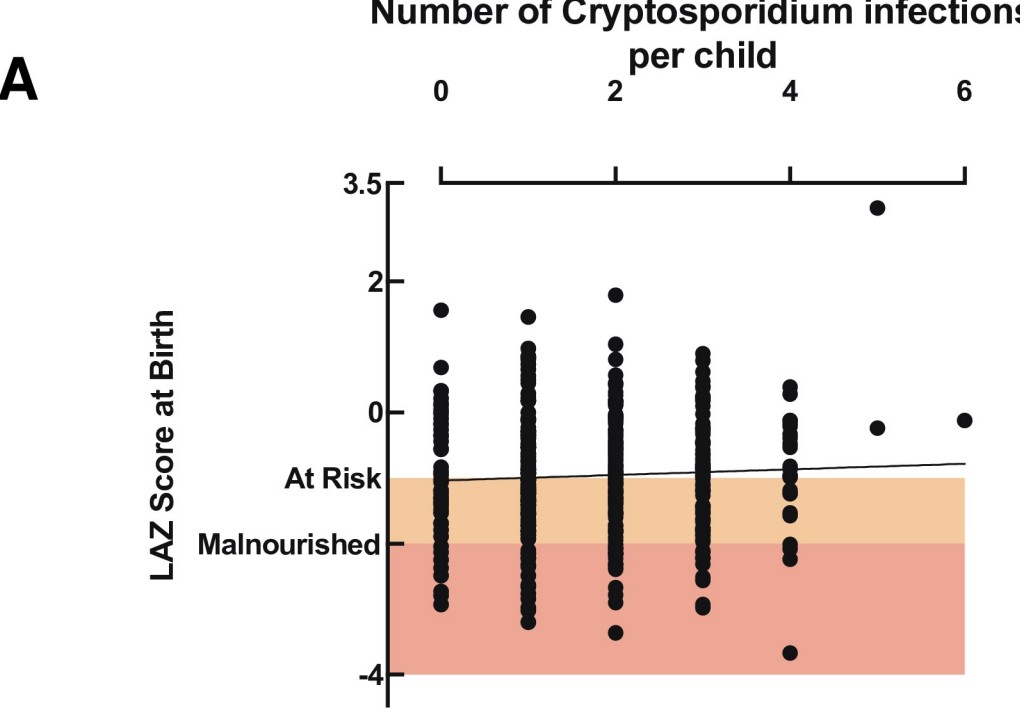

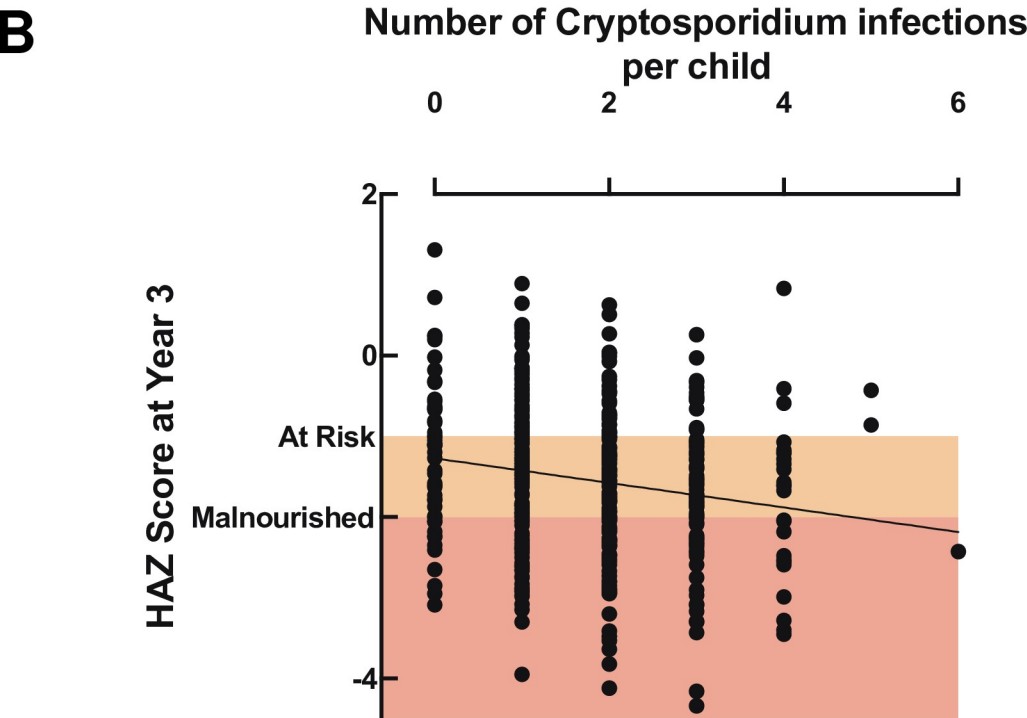

**Fig 4. Cryptosporidiosis frequency was associated with growth faltering distinct from the impact of birth nutritional status.** A) Relationship between length for age z score (LAZ) at birth (Y-axis) and the total number of *Cryptosporidium*

*spp*. infections (X-axis). The slope was not significantly different from one. B) Relationship between height for age z score (HAZ) at 3 years (Y-axis) and the total number of *Cryptosporidium spp*. infections (X-axis). This was initially analyzed by use of linear regression Slope: -0.152 ± 0.0429, R squared value: 0.0313, Significance p=0.0004 and the relationship between the two data sets was then confirmed by the Pearsons correlation coefficient as described in the text. Children were defined to be at risk for growth faltering with a LAZ or HAZ score <-1 and malnourished at LAZ or HAZ score <-2. Orange box: birth LAZ or 3 year HAZ score -1 to -2; red box: birth LAZ or 3 year HAZ or score < -2.

(Fig 6). Antibodies against Cp23 are not however always developed in response to a *Cryptosporidium* infection. In addition to the analysis shown in Fig 6 the data was reexamined to determine if children exposed to the parasite but who did not have an antigenic response experienced less growth faltering after a *Cryptosporidium* infection than immunologically naive children. In S7 Fig the children were subgrouped into children who had been not exposed to the parasite, infected but had not developed a strong IgA response (as determined by the detection of *Cryptosporidium* DNA by qPCR) (representative examples S8A, S8B and S8C Fig), uninfected <365 days (representative example S8D Fig) or children who had high levels of IgA (representative examples S8E and S8F Fig). There was no difference between the birth LAZ of these different groups. Children infected with the *Cryptosporidium* parasite but who had low levels of fecal anti-Cp23 IgA had lower HAZ scores than children with high levels of fecal anti-Cp23 IgA (S7A Fig). The same result was obtained if the analysis was performed using the fecal IgA antibodies against a second sporozoite peptide (Cp17) to define the subgroups (S7B Fig).

## Discussion

The key finding of this paper is that naturally acquired immunity protects from *Cryptosporidium* diarrhea but does not provide sterilizing immunity. The importance of this observation is two-fold: first it indicates that transmission likely occurs in semi-immune populations; and second that continued sub-clinical infections increase the risk of infection-related growth faltering. Encouragingly however, acquired immunity associated with high levels of mucosal IgA against the Cp23 cryptosporidium sporozoite antigen were associated with protection from malnutrition.

Many previous studies on cryptosporidiosis have focused on the health impact of diarrhea-associated cryptosporidiosis [7,12,29–32]. However sub-clinical disease, as opposed to infection accompanied by diarrhea, may also have long term effects on child health. The link between sub-clinical cryptosporidiosis and malnutrition is now well known if not yet well understood [8,9,13,30]. In a recent study the global prevalence of cryptosporidiosis in people without diarrheal symptoms was 4.4% (95% confidence interval 2.9–6.3)[33]. During the 3 years of this study 212 children (54%) had only sub-clinical *Cryptosporidium* infections. This longitudinal study allowed us to take an in depth look at the role of sub-clinical reinfections in the exacerbation of growth faltering [13,21,24].

Anthropometric measurements are reliable non-invasive methods to monitor child malnutrition. The most commonly used metrics are a shortfall in child growth (low height for age: HAZ score) a consequence of chronic undernutrition and wasting (exemplified by a low weight for height: WHZ score). In line with most studies our results show that a history of cryptosporidiosis was associated with a decrease in the HAZ score of children irrespective of infection severity [8,25,34,35]. Here we found that child growth was negatively impacted not only by the first episode of cryptosporidiosis, but both occurred and remained constant in succeeding infections, even though parasite burden and diarrheal disease decreased. This study has, therefore, shown that naturally acquired partial immunity was not effective at preventing growth faltering and that a control strategy focused on only preventing diarrheal cryptosporidiosis may not prevent the stunted growth associated with cryptosporidiosis.

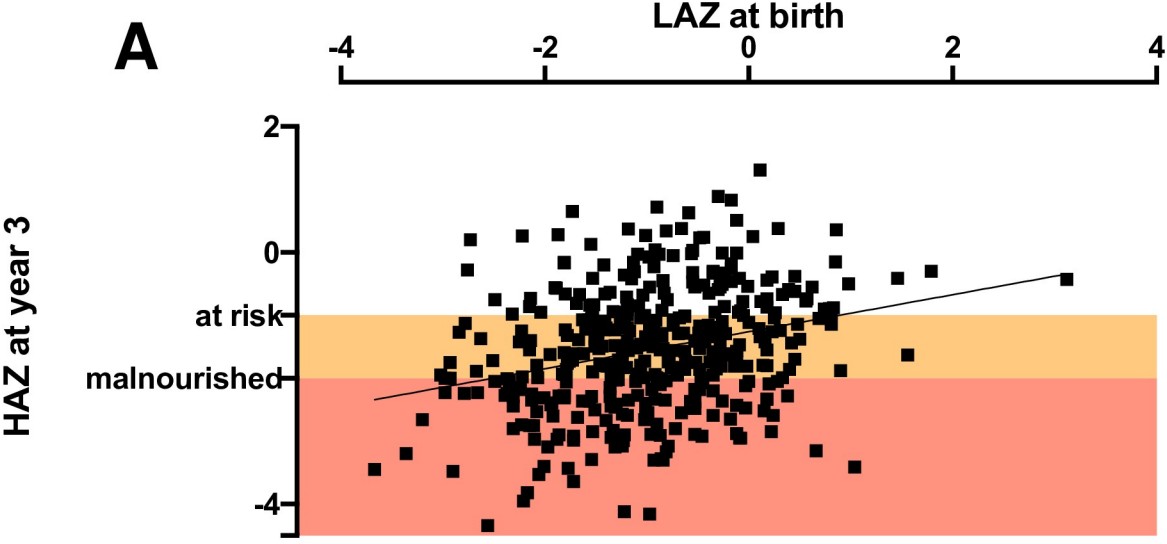

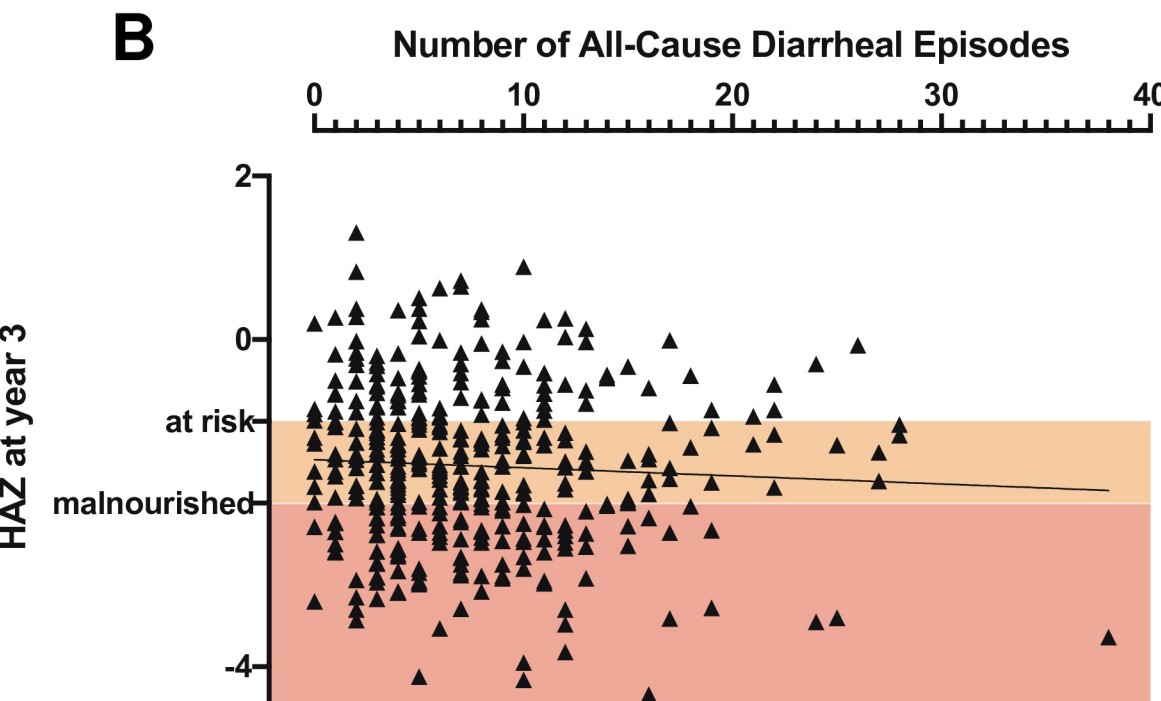

**Fig 5. Correlates of cryptosporidiosis-associated growth-faltering.** A) Comparison of three year-HAZ with birth LAZ. (Slope: -0.294 ± 0.05; R squared value: 0.08; Significance p < 0.0001). B) Relationship of all-cause diarrhea with HAZ at 3 years of age (p = NS).

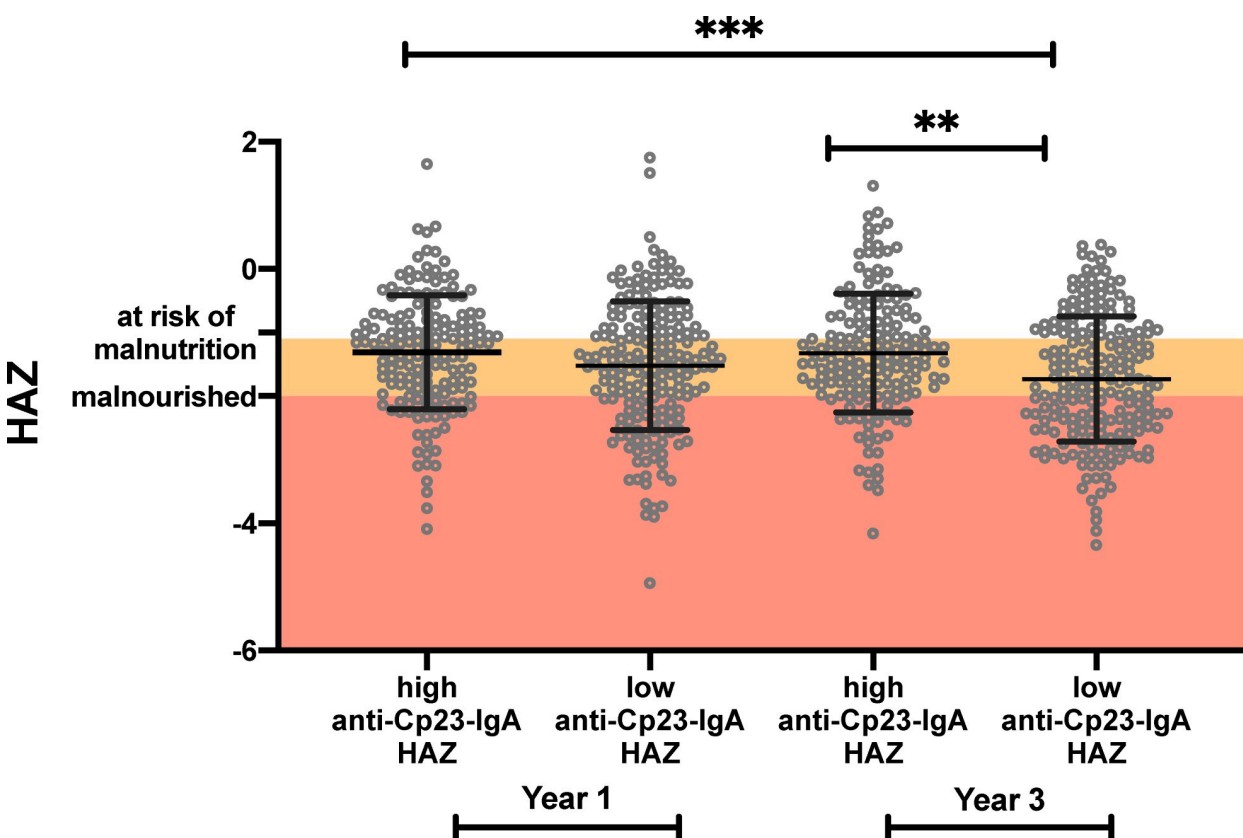

**Fig 6. High anti-Cp23 IgA levels were associated with a reduction in cryptosporidiosis-associated growth-faltering.** The HAZ values of children in the upper and lower 50th percentile for fecal IgA anti-Cp23 year were analyzed. HAZ are shown for children in both year one and year three of life. Mean ± standard deviation with individual data points. Horizontal bars represent the result of a non-parametric Kruskal-Wallis test ***p<0.001, **p<0.01.

A limitation of the current study is that it was not possible to unambiguously attribute an episode of diarrhea to *Cryptosporidium* because children in this community were infected with multiple enteropathogens at the same time [25,28]. To mitigate the problem of correctly identifying *Cryptosporidium*-associated diarrheal infections these were defined as an episode of diarrhea accompanied by a new *Cryptosporidium* infection (i.e. the immediately preceding surveillance or diarrheal stool sample was negative for *Cryptosporidium*) [13]. A second limitation was that surveillance stool samples were collected at only monthly intervals which likely missed some subclinical infections, potentially underestimating the impact of cryptosporidiosis on child growth. The study however had notable strengths including most importantly its longitudinal design that combined collection of surveillance and clinical specimens with studies on child growth faltering.

The association of mucosal immunity to Cp23 with protection from growth faltering offers hope that a cryptosporidiosis vaccine could have a measurable impact on child health, even in the absence of absolute protection from infection.

## Methods

### Ethics statement

The study was approved by the Ethical and Research Review Committees of the International Centre for Diarrhoeal Disease Research, Bangladesh (PR-13092) and by the Institutional

**Table 4. Maternal and family demographics.**

| Maternal and Family Characteristics | N=392 |
|---|---|
| Mean Maternal Age, year (SD) | 24.63 (4.68) |
| Mean Maternal Weight, Kg (SD) | 51.82 (10.17) |
| Mean Maternal Height, Cm (SD) | 149.71 (5.03) |
| Mean Maternal BMI, kg/m2 (SD) | 23.10 (4.32) |
| No Maternal Education, N (%) | 87 (22.2) |
| Median Household income (BDT*) (IQR) | 14,000 (10,000) |
| Treated water (Boil), N (%) | 390 (74.0) |

**Abbreviations**: SD, Standard Deviation; BDT, Bangladesh Taka; IQR, Inter Quartile Range

*1000 BDT is approximately 12 US dollars

Review Board of the University of Virginia (IRB#20388). Informed written consent was obtained from the parents or guardians for the participation of the subjects in the study.

## Child cohort

A total of 500 children were enrolled within one week of birth in an urban slum of Dhaka, Bangladesh beginning in June 2014 through March 2016 and were monitored for diarrheal diseases through bi-weekly home visits by trained field investigators. A monthly stool sample was also collected to evaluate asymptomatic infection and growth was measured every 3 months ("Cryptosporidiosis and Enteropathogens in Bangladesh"; ClinicalTrials.gov identifier NCT02764918). This area (Section 11 of Mirpur Thana) is densely populated with participants in this study having an average of 5.5 people living in 1.6 rooms. Annual median household income of participants was 14,000 Taka or approximately US $164 (Table 4). Anthropometric data was collected as previously described [13]. Each child was weighed on an electronic scale (kilograms, measured with electronic scale; TANITA, HD-314). Child height or length (depending on age) and mid-upper arm circumference were measured to the nearest 0.1 cm using a measuring board and plastic tape (Table 5). The height-for-age z score (HAZ); weight for age z score (WAZ); weight for height (WHZ); body mass index for age (BAZ); and mid-upper arm circumference for age (MUACZ) were calculated using the World Health Organization Anthro software (version 3.2.2) [13]. Children who had a HAZ score <-1 were defined

**Table 5. Infant demographic characteristics.**

| Infant Demographic Characteristics | Year 2 N=421 | Year 3 N=392 |
|---|---|---|
| Gender, Female N (%) | 229 (54.4) | 214 (54.6) |
| Mean Infant Age in days, Range | 733 (719–783) | 1099 (1035–1134) |
| Mean Weight, Kg, (SD) | 10.17 (1.31) | 11.93 (1.52) |
| Mean Height, Cm, (SD) | 81.72 (3.18) | 89.78 (3.74) |
| Mean MUAC, Cm, (SD) | 14.87 (1.02) | 15.3 (1.00) |
| Mean WAZ, (SD) | -1.34 (1.05) | -1.42 (0.99) |
| Mean HAZ, (SD) | -1.56 (0.98) | -1.54 (0.98) |
| Mean MUACZ, (SD) | -0.15 (0.85) | -0.34 (0.79) |
| Mean BAZ, (SD) | -0.52 (0.98) | -0.63 (0.91) |

**Abbreviations**: SD, Standard Deviation, MUAC, Mid Upper Arm Circumference, WAZ, Weight-for-age, HAZ, Height-for-age, MUACZ, Mid Upper Arm Circumference-for-age, BAZ, body mass index-for-age

as 'at risk for malnutrition' and HAZ < -2 as malnourished [26,36]. Diarrhea was defined as ≥3 loose stools within a 24-hour period as reported by the child's caregiver with episodes separated by a gap of at least 3 days. This paper reports the data from 392 infants who were followed through three years of age.

## Sampling and specimen testing

Fresh stool samples collected in the field were placed on ice and then brought to the lab on the same day and frozen within 6 h of collection (S1 Fig). Stool specimens were collected from children every month (monthly surveillance) and during episodes of diarrhea. A modified Qiagen stool DNA extraction protocol with 95˚C incubation and a 3-minutes bead–beating step was used to extract DNA [13] (S9 Fig). These samples were tested with a multiplex qPCR assay which utilizes pan-Cryptosporidium primers and probes targeting the 18S rDNA gene and primers and probes to detect the Phocine herpesvirus (PhHV) extraction control (obtained from the European Virus Archive Global organization) as previously described (S10 Fig). All samples with a cycle threshold of ≤ 40 for cryptosporidium were used in this analysis [13]. In year 3 the diagnostic qPCR assay was not able to be completed on 0.9% of the collected diarrheal and 4.6% of the monthly surveillance samples (S2 Table).

Infection with *Cryptosporidium* was defined as detection of *Cryptosporidium* DNA by qPCR from stool. PCR- positive samples were classified as a separate infection if occurring greater than 65 days after the preceding positive sample [13]. The *Cryptosporidium* infection phenotype (diarrheal or sub-clinical) was based upon symptoms at the time of detection of the first *Cryptosporidium* - positive stool sample, whether diarrheal stool or monthly surveillance.

## Statistical analysis

Descriptive statistics were expressed in mean ± standard deviation for continuous variables and as frequencies and proportions for categorical variables. The frequency of repeated *Cryptosporidium* infections in the first 3 years of life was summarized for diarrhea and sub-clinical infections separately and their differences were evaluated with the $\chi^2$ test. To account for within-child correlations among repeated *Cryptosporidium* infections, the relationship between parasite burden and the number of repeated *Cryptosporidium* infections was evaluated using the Generalized Estimating Equation (GEE) for repeated measurements, assuming an exchangeable correlation structure. Pearson correlation was calculated for univariate association of individual predictors with HAZ at 3 years. Since confounders such as LAZ at birth, maternal weight and height, maternal education, household income and access to treated water were previously shown to impact HAZ [8,13], a multivariable linear regression was performed to evaluate the association between *Cryptosporidium* infection and HAZ at 3 years after adjusting for these factors (Table 3). Similarly, a multiple regression analysis was performed to independently evaluate whether the number of episodes of diarrhea, irrespective of the causative pathogen, was associated with HAZ at 3 years. Analyses were performed using both the GraphPad Prism version 8.4.3 for Mac, (GraphPad Software, San Diego, California USA,), SAS 9.4 (Raleigh, NC) and R version 3.3.3, 32-bit.

## Supporting information

**S1 Table. Multivariable analysis of total all cause diarrhea and HAZ at year 3.**
(TIF)

**S2 Table. Symptomatic and asymptomatic samples collected during year 3.**
(TIF)

**S1 Fig. Flow chart of stool processing and molecular testing.**
(TIF)

**S2 Fig. Parasite Burden in diarrheal and sub-clinical infections.** Relationship between Parasite Burden and the number of recurrent *Cryptosporidium* infections. Each symbol represents the first detectable sample of an individual infection. Y-axis, quantitative cycle of the diagnostic pan-Cryptosporidium PCR assay (Cq). X-axis, the total number of *Cryptosporidium* infections. The infection was designated as either diarrheal (red) or sub-clinical (green) based on the current infection phenotype. The data from diarrheal cases was offset to improve data visualization. To account for within-child correlations among repeated *Cryptosporidium* infections, the generalized estimating equation (GEE) method for repeated measurements were used with exchangeable correlation structure. As the intercept of the diarrheal and sub-clinical models was not statistically different the common intercept (27.02 ± 0.45) was used. The slope of the data derived from the sub-clinical (1.9 ± 0.2) and diarrheal (1.49 ± 0.31) exchangeable models were not significantly different from each other (p=0.071) although both were statistically different from zero (p<0.0001).
(TIF)

**S3 Fig. Distribution of repeated *Cryptosporidium* infections.** x axis child age in months; left y-axis child HAZ scores; right y-axis frequency of *Cryptosporidium* (diarrheal and sub-clinical) infections (shown as the number that occurred per the age of the child in months). All graphs include as a reference the HAZ score of children where no *Cryptosporidium* infections were detected (green circle and line). *Cryptosporidium* infections: Light blue triangle dotted blue connection line: infection one; purple circle and dotted line: infection two; light red square and dotted line: infection three; black triangle and dotted line: infection four A) blue symbol and solid line HAZ score of children who had one *Cryptosporidium* infections by 3 years of age B) purple square and solid line HAZ score of children who had two *Cryptosporidium* infections by three years of age C) red square and solid line HAZ score of children who had three *Cryptosporidium* infections by three years of age D) black square and solid line HAZ score of children who had four *Cryptosporidium* infections by three years of age.
(TIF)

**S4 Fig. Recurrent cryptosporidiosis results in greater growth faltering Each symbol represents a single child.** Box plot comparing the height for age z score at 3 years (HAZ) (Y-axis) mean and standard deviation shown Children were considered to be at Risk for malnutrition is they have a HAZ score <-1 and malnourished at HAZ-2: orange box: 3-year HAZ score -1 to -2; red box 3-year HAZ score < -2. X-axis Number of Cryptosporidium infections. Bar indicates the result of a non-parametric Kruskal-Wallis test for multiple comparisons * indicates p<0.05 ** indicates p<0.01.
(TIF)

**S5 Fig. Comparison of Cryptosporidiosis associated Growth-faltering in diarrheal and sub-clinical infections.** Graphs show results from a simple linear regression with each symbol representing a single child. Black symbols represent children who were never infected or had sub-clinical infections. The blue symbols indicate children who have had one or more than one episodes of diarrhea- associated cryptosporidiosis. Height for age (HAZ) z score at 3 years is shown on the Y-axis. The slope of the diarrheal-associated and sub-clinical groups are identical. Pooled Slope: -0.1545. Children are considered to be at Risk for malnutrition if they have a HAZ score <-1 and malnourished at HAZ -2: orange box: 3-year HAZ score -1 to -2; red box: 3-year HAZ score < -2. X-axis indicates number of *Cryptosporidium* infecti**ons**.
(TIF)

**S6 Fig. Cryptosporidiosis was associated with chronic but not acute malnutrition at year 3.** Graphs show results from a simple linear regression with each symbol representing a single child X-axis indicates number of *Cryptosporidium* infections A) Y-axis MUACZ circumference of the mid-upper arm (muscle wasting) B) Y-axis WHZ score (low weight for height (wasting) a measure of acute malnutrition C) Y axis WAZ score (low weight for age) a measure of acute and chronic malnutrition and D) BAZ (body mass index for age).
(TIF)

**S7 Fig. Low anti-Cryptosporidium IgA levels after an infection were associated with a subsequent increase in cryptosporidiosis-associated growth-faltering.**
(TIF)

**S8 Fig. Parasite Burden in Reperesentative children.** Each symbol represents the first detectable sample of an individual infection. Y-axis, quantitative cycle of the diagnostic pan-Cryptosporidium PCR assay (Cq). X-axis age of child in days A-C Infants with high fecal IgA anti-Cryptosporidium at year one D-F Infants with low fecal IgA anti-Cryptosporidium at year one G).
(TIF)

**S9 Fig. Flow chart of stool TNA extraction procedure using QIAamp Fast DNA Stool Mini Kit from fresh or frozen stool samples.**
(TIF)

**S10 Fig. Flow chart of Multiplex qPCR of Cryptosporidium, Giardia, Entameoba histolytica by targeting the18S gene.**
(TIF)

## Acknowledgments

We are grateful for the participation of the parents and children in this study as well as the staff of the Emerging Infectious Diseases Division of icddr,b for contributing to this research. The governments of Bangladesh, Canada, Sweden, and the UK provide core support to icddr,b.

## Author Contributions

**Conceptualization:** Mamun Kabir, Rashidul Haque, William A. Petri, Jr, Carol A. Gilchrist.

**Data curation:** Masud Alam, Uma Nayak.

**Formal analysis:** Uma Nayak, Jennie Z. Ma, Carol A. Gilchrist.

**Funding acquisition:** William A. Petri, Jr, Carol A. Gilchrist.

**Investigation:** Tuhinur Arju, Biplob Hossain, Rubaiya Tarannum, Amena Khatun.

**Methodology:** Masud Alam, Jennifer A. White, Rashidul Haque, William A. Petri, Jr, Carol A. Gilchrist.

**Project administration:** Masud Alam, Jennifer A. White, Rashidul Haque, William A. Petri, Jr, Carol A. Gilchrist.

**Resources:** Rashidul Haque, William A. Petri, Jr.

**Supervision:** William A. Petri, Jr.

**Writing – original draft:** Mamun Kabir, Carol A. Gilchrist.

**Writing – review & editing:** Masud Alam, Uma Nayak, Tuhinur Arju, Biplob Hossain, Rubaiya Tarannum, Amena Khatun, Jennifer A. White, Jennie Z. Ma, Rashidul Haque, William A. Petri, Jr, Carol A. Gilchrist.

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
