## [Decision Letter · Decision Letter 0]

31 Mar 2021

Dear Dr. Petri, Jr.,

Thank you very much for submitting your manuscript "Nonsterile immunity to cryptosporidiosis in infants is associated with mucosal IgA against the sporozoite and protection from malnutrition" for consideration at PLOS Pathogens. As with all papers reviewed by the journal, your manuscript was reviewed by members of the editorial board and by several independent reviewers. The reviewers appreciated the attention to an important topic. Based on the reviews, we are likely to accept this manuscript for publication, providing that you modify the manuscript according to the review recommendations.

Please pay particular attention to the comments from reviewer 3, who raises some important questions about the link between immunity and infections as well as parasite burden and outcomes. Please note that the journal has expanded its scope and welcomes manuscripts with a clinical focus, recognizing that studies in humans may by their nature be correlative rather than purely mechanistic. 

Sincerely,

Kami Kim

Section Editor

PLOS Pathogens

Kami Kim

Section Editor

PLOS Pathogens

Kasturi Haldar

Editor-in-Chief

PLOS Pathogens

orcid.org/0000-0001-5065-158X

Michael Malim

Editor-in-Chief

PLOS Pathogens

orcid.org/0000-0002-7699-2064

Reviewer Comments (if any, and for reference):

Reviewer's Responses to Questions

**Part I - Summary**

Reviewer #1: This manuscript by Kabir and colleagues is focused on immunity to cryptosporidiosis in children. Methodologically this paper is somewhat outside the ‘typical’ PLoS Pathogens focus on lab-based studies that aim at molecular mechanism. However, the editor sent this for review, and thus likely made the decision that it is well within the scope of the journal. This manuscript is longitudinal observational study in a cohort of young children that grow up in an environment of very high likelihood of exposure to Cryptosporidium infection. From a public health perspective this is by far the most relevant group. Several studies including GEMS and MALED have previously shown significant correlation of cryptosporidiosis with diarrheal disease, malnutrition, stunting, and death. The introduction (and discussion) is rather brief on this important aspect and could provide more relevant background to the reader – in particular for this journal and audience. Also given the audience a bit more information on the biology of the pathogen may be helpful. Whether, and to which extend, humans develop immunity to this infection is an important and not fully settled question with obvious impact on the probability of vaccine success. This study addresses several important issues. The key finding is that while most children experience multiple infections the parasite and disease burden decreases significantly along the timeline which the authors interpret as non-sterile disease burden. Measurements of antibody titers are a reasonable independent correlate and support the main conclusion. The true value of the study lies in the size of the group of children, the excellent retention and the high frequency of measurements regardless, whether children showed signs of disease or not. This is a significant accomplishment and a unique resource to the field. The question whether kids gain resistance over time due to immunity or due to age, or age associated confounders like changes in food or microbiome, has been debated for some time. The study group here is of sufficient size to control for age and importantly finds the protective effect to be age independent. The study also uses a comparison of single infection to first in several as a clever way to control for overall differences in susceptibility to the infection. As other studies they find significant association between infection with malnutrition and growth faltering and their longitudinal data allow better treatment of some of the confounders, and importantly, they find that overt disease is not required to cause this. Interestingly, cryptosporidiosis (symptomatic or not) was a stronger predictor than all cause diarrheal disease. Overall, I read the paper with interest and found the main conclusions to be important and well supported and the discussion balanced. For the broader audience the discussion could have featured comparison to other infections. Rotavirus may be of particular interest here as for this virus, similar to what the authors describe for an apicomplexan here, immunity is non-sterile but associated with significant benefit. Also, discussion of the literature around the rotavirus vaccine may offer the authors opportunity to critically discuss the potential value of such prevention for cryptosporidiosis.

Reviewer #2: This manuscript reports the results of a birth cohort study that prospectively studied Cryptosporidium infection (by qPCR) both symptomatic and asymptomatic and nutrition. Overall, the study demonstrates immunity against diarrhea with repeat infections, but not sterile immunity. Also, there was a clear effect of both diarrhea and non-diarrheal infections on measures of chronic malnutrition in children. The authors also noted an association between a marked of immunity (fecal IgA versus Cp23) and protection from the nutritional effects of reinfection, despite the absence of sterile immunity. These are important observations and the data seem to be quite rigorous, though observational and not mechanistic.

Reviewer #3: This manuscript describes a well-designed birth cohort study to look at the effects of cryptosporidiosis and immunity to cryptosporidiosis on child development. The results show that as the number of repeat infections increase, there is less diarrhea associated with infection and a lower parasite burden (as determined by Cq values of the first positive sample of each infection). Growth faltering is worse as the number of repeat infections increase. The authors then show that high levels of fecal anti-Cp23 IgA in the first year of life is associated with protection from growth faltering through year 3. It is critical for the authors to highlight how their results add to previously published data on cryptosporidiosis, immunity to cryptosporidiosis, and growth faltering; most of what they report has been reported from other studies. The data as presented leaves many questions. For example, how do IgA levels relate to repeat infections? Presumably, immunity increases with repeat infections (suggested by lower parasite burdens and less diarrhea) but children with repeat infections have worse outcomes. Do the children with high IgA in Figure 6 (Group 1) also have fewer infections? What is the parasite burden for each child over the 3 years? Does this correlate with outcomes?

**Part II – Major Issues: Key Experiments Required for Acceptance**

Reviewer #1: (No Response)

Reviewer #2: None

Reviewer #3: The authors state in the discussion that “The key finding of this paper is that naturally acquired immunity protects from Cryptosporidium diarrhea but does not provide sterilizing immunity”, but the results do not show that increased immunity leads to less diarrhea-all that is shown is that increased immunity leads to less growth faltering. Can the relationship between IgA levels and repeat infections be analyzed and described?

In the analysis of the samples, parasite burden has been quantified by qPCR, but only the Cq of the first positive sample is used in the analysis. Is there any way to use the qPCR results to estimate total parasite burden over the 3 years?

It is not stated in the methods the timing of sample collection during diarrheal episodes-were diarrheal samples collected and tested daily? Is it possible to quantify parasite burden of a diarrheal episode by area under the curve analysis?

**Part III – Minor Issues: Editorial and Data Presentation Modifications**

Reviewer #1: Minor comments and corrects:

The introduction is rather brief.

The authors might briefly specify the definition of a repeat infection in the results section (it is well described in the M&M) as this is an important feature of the experimental set up.

The investigators use value of the first detectable sample as the ‘amplitude’ of burden. Why did they do that and not something like area under the curve?

I feel it may help the reader to understand this infection and the study design and results better, to see several ‘representative’ children over the entire time with all samples, essentially cv burden curves over the three years highlighting examples of the ‘epidemiological phenotypes’ they discuss later. Either in fig. 1 or as a supplementary figure.

L58 cut ‘were’

Reviewer #2: Table 2. This table is not very clear and overall distracts from the points. I think the manuscript would be stronger with it deleted.

Fig 3 and Fig S4. Some of the. points noted have a cQ of 40, which barely meets the case definition. Please check that these were in fact positive.

Fig 5. This figure duplicates data in the top and bottom half of the figure. Both are not needed.

Fig 6 would be clearer if labeled IgA high and low rather than group 1 and 2.

Reviewer #3: Fig 1: please clarify “within 65 days of previous”

Table 2 does not show Cq values, but the way the text is worded in lines 93-94 makes it sound like the table does have Cq values. Please re-word to make the text clearer.

In many figures the mean and error bars cannot be seen (ie, Fig 2B, Fig S6 and Fig 6). Please use a different color for the bars.

Fig 5A. Is the bar to the right of the table supposed to be color coded?

Figure S5 A-D need to be enlarged.

Line 162-Should the referral to Table 5 be instead a referral to Figure S5?

In lines 186-187 the text states the Pearson coefficient and p value for the Cryptosporidium infections: HAZ at year 3 comparison as: coef = -0.18, p=0.024. But in lines 193 to 195 the text states: ”However, while Cryptosporidium infections (diarrheal and sub-clinical) were significantly associated with child HAZ at year 3 (Pearson’s correlation p=0.0004). These statements seem to report different p values for the same comparison. Please clarify.

Line 196: Should say Table S2.

Lines 211-214 and Figure S9A. Can you please clarify the reason why Group 2 at year 1 is split out into uninfected (2a) and pcr+ (2b) children? What does “diagnostic qPCR positive only” mean? It is unclear what children are included in group 2b. Was infection diagnosed by other methods?

PLOS authors have the option to publish the peer review history of their article (what does this mean?). If published, this will include your full peer review and any attached files.

Reviewer #1: No

Reviewer #2: No

Reviewer #3: No

Figure Files:

Data Requirements:

Reproducibility:

References:

---

## [Editor Report · Decision Letter 1]

16 May 2021

Dear Dr. Petri, Jr.,

We are pleased to inform you that your manuscript 'Nonsterile immunity to cryptosporidiosis in infants is associated with mucosal IgA against the sporozoite and protection from malnutrition' has been provisionally accepted for publication in PLOS Pathogens.

Best regards,

Kami Kim

Section Editor

PLOS Pathogens

Kami Kim

Section Editor

PLOS Pathogens

Kasturi Haldar

Editor-in-Chief

PLOS Pathogens

orcid.org/0000-0001-5065-158X

Michael Malim

Editor-in-Chief

PLOS Pathogens

orcid.org/0000-0002-7699-2064
---

## [Editor Report · Acceptance letter]

21 Jun 2021

Dear Dr. Petri, Jr.,

We are delighted to inform you that your manuscript, "Nonsterile immunity to cryptosporidiosis in infants is associated with mucosal IgA against the sporozoite and protection from malnutrition," has been formally accepted for publication in PLOS Pathogens.

Best regards,

Kasturi Haldar

Editor-in-Chief

PLOS Pathogens

orcid.org/0000-0001-5065-158X

Michael Malim

Editor-in-Chief

PLOS Pathogens

orcid.org/0000-0002-7699-2064